# Characterization of Naturally Occurring Bioactive Factor Mixtures for Bone Regeneration

**DOI:** 10.3390/ijms21041412

**Published:** 2020-02-19

**Authors:** Henriette Bretschneider, Mandy Quade, Anja Lode, Michael Gelinsky, Stefan Rammelt, Stefan Zwingenberger, Klaus-Dieter Schaser, Corina Vater

**Affiliations:** 1University Center of Orthopaedics and Traumatology, University Hospital Carl Gustav Carus of Technische, Universität Dresden, Fetscherstraße 74, 01307 Dresden, Germany; 2Centre for Translational Bone, Joint and Soft Tissue Research, University Hospital Carl Gustav Carus and Faculty of Medicine of Technische, Universität Dresden, Fetscherstraße 74, 01307 Dresden, Germany

**Keywords:** growth factor, hypoxia-conditioned medium, platelet lysates, adipose tissue extract, bone regeneration

## Abstract

In this study, the bone-regenerative potential of bioactive factors derived from adipose tissue, platelet-rich plasma (PRP) and conditioned medium from hypoxia-treated human telomerase immortalized bone-marrow-derived mesenchymal stem cells (hTERT-MSC) was investigated in vitro with the aim to develop cost-effective and efficient bone substitutes for optimized regeneration of bone defects. Adipose tissue was harvested from human donors undergoing reconstructive surgery, and adipose tissue extract (ATE) was prepared. Platelet lysates (PL) were produced by repeated freeze-thaw cycles of PRP, and hypoxia-conditioned medium (HCM) was obtained by culturing human telomerase immortalized bone-marrow-derived mesenchymal stromal cells for 5 days with 1% O2. Besides analysis by cytokine and angiogenesis arrays, ELISA was performed. Angiogenic potential was investigated in cocultures of bone-marrow-derived (BM)-MSC and human umbilical vein endothelial cells. Multiple angiogenic proteins and cytokines were detected in all growth factor mixtures. HCM and ATE contained high amounts of angiogenin and CCL2/MCP-1, whereas PL contained high amounts of IGFBP-1. Culturing cells with HCM and ATE significantly increased specific ALP activity of BM-MSC as well as tubule length and junctions of endothelial networks, indicating osteogenic and angiogenic stimulation. To achieve a synergism between chemoattractive potential and osteogenic and angiogenic differentiation capacity, a combination of different growth factors appears promising for potential clinical applications.

## 1. Introduction

Local bone loss that may arise from trauma, tumor, infection, periprosthetic osteolysis or congenital musculoskeletal disorders constitutes a major worldwide socioeconomic problem frequently requiring surgical intervention. Although bone autografts are the gold standard for filling the defect, the available amount of autologous bone is limited, and its harvesting is associated with several drawbacks like the additional donor-site morbidity [1]. Allo- or xenografts carry the risk of immunogenic rejection and the potential for disease transmission [2,3]. Artificial matrix materials based on e.g., calcium phosphates, bioglass, synthetic or biological polymers and composites were developed to overcome the drawbacks of auto-, allo- and xenografts [4,5,6,7,8,9]. However, for bone regeneration, the ingrowth of osteogenic cells, as well as blood vessels to ensure sufficient nutrient and oxygen supply, is important. With the aim of obtaining an osteoinductive and angiogenic effect, these materials are combined with growth factors and/or cells [10,11]. Simultaneous delivery of osteogenic and angiogenic factors may therefore significantly enhance the success of bone repair therapies [10]. 

Bone formation in vivo is a sequential multistep process comprising activation, chemotaxis, mitosis and differentiation of cells and is regulated by a large number of interacting factors. Therefore, combinations of growth factors are more efficient for tissue regeneration applications than the use of single ones [12,13,14,15]. Instead of using several recombinant growth factors in high concentrations, which may lead to serious side effects (e.g., ectopic bone formation, osteolysis), different natural sources of bioactive factors are proposed and investigated [16,17,18].

Platelet-rich plasma (PRP) is an abundant source of growth factors that has beneficial effects on angiogenesis and wound and bone healing [19,20,21,22]. Furthermore, it has been shown to improve bone formation within tissue-engineered constructs [21]. Autologous PRP has been used in combination with expanded bone marrow cells for the treatment of long bone defects [23]. 

Conditioned medium from hypoxia-treated (HCM) bone-marrow-derived mesenchymal stromal cells (BM-MSC) contains growth factors such as vascular endothelial growth factor (VEGF) and high-mobility group protein B1 (HMGB1), leading to a high attraction of BM-MSC [24]. Due to hypoxic cultivation conditions, their concentration of these factors in the cell culture medium is significantly higher than in medium from normoxic conditions [25]. Other groups also reported HCM as a source of paracrine factors with the potential to enhance migration, proliferation and vascularization [26]. In addition, a high potential for induction of bone regeneration by conditioned medium from MSC, even under normoxic conditions, has been reported for a bone defect model in vivo [27]. 

Adipose tissue extract (ATE) derived from lipoaspirates is another natural source of bioactive factors [28]. Adipose tissue is the largest endocrine organ of the body and secretes high amounts of growth- and differentiation-supporting factors (e.g., leptin, adiponectin, angiotensin, glucocorticoids, interleukins, growth factors) which can be used for therapeutic applications [29,30,31]. Some of these factors stimulate angiogenesis and the formation of extended capillary networks. ATE-functionalized hydrogels induced e.g., adipogenic differentiation of adipose tissue-derived stem cells in vitro, showed a sustained release of these factors and stimulated angiogenesis in vivo [32].

The purpose of this study was to comparatively evaluate the bone-regenerative and angiogenic potential of naturally occurring bioactive factors derived from adipose tissue, hypoxia-conditioned cell culture supernatants and platelets with the aim to develop efficient and cost-effective active agents to functionalize bone substitutes for enhanced regeneration of large bone defects.

## 2. Results

### 2.1. Protein Quantification, Angiogenesis Protein and Cytokine Array

Mean protein concentrations were measured as shown in Table 1. Platelet lysates (PL) had the highest protein concentration - approximately 770 times higher than HCM. Using an angiogenesis and cytokine array, a specific protein profile of the growth factor mixtures was obtained (Figure 1 and detailed in Appendix A). With respect to the protein analysis, angiogenin was the most abundant protein in HCM and ATE, whereas in PL, amounts of IGFBP-1 were highest. PDGF-AA could be detected in PL and ATE, but not in HCM. IL-6 was only quantifiable in HCM and ATE. As determined by ELISA, TIMP-1 as angiogenesis suppressor was abundant in all three growth factor mixtures (Table 2). Regarding pro-angiogenic proteins, high concentrations of VEGF and angiogenin could be detected. With respect to the protein array, VEGF was highest in ATE, but in terms of ELISA, highest in HCM (Table 2). For TIMP-1, PDGF-BB, IGFBP-1, angiogenin, IL-1ß and IL-6, the proportions were the same when comparing protein array and ELISA data. Besides angiogenin, the percentages for CXCL-1, bFGF and IL-8 differed between the array and ELISA results.

### 2.2. Chemoattractive Potential, Proliferation and Specific ALP Activity 

Compared to the negative control (medium with 0% fetal calf serum (FCS)), PL displayed a significantly higher chemoattractive potential towards BM-MSC, similar to the positive control, and a significantly higher chemoattractive potential than ATE and HCM (30% FCS, *p* < 0.001; Figure 2). 

Compared to the positive control (ctrl + osteogenic supplements (OS)), culture medium containing 10% FCS), addition of 10% of all three growth factor mixtures to the medium (culture medium contained no additional FCS) led to a significantly decreased proliferation of BM-MSC during a 14-day cultivation period (Figure 3A). HCM + OS and ATE + OS were significantly inferior to PL + OS in this respect. In contrast, a significantly higher specific ALP activity was detected after addition of HCM (*p* < 0.0001) and ATE (*p* < 0.0021) as compared to the positive control group (ctrl + OS; Figure 3B). In general, osteogenic differentiation could only be observed when growth factor mixtures were added in combination with osteogenic supplements (Dex, AAP, β-GP; data of experiments without OS not shown).

### 2.3. Angiogenic Potential of PL, HCM and ATE 

The angiogenic potential of the three different growth factor mixtures was compared by culturing BM-MSC in co-culture with HUVEC for up to 10 days in media containing PL, HCM or ATE. After CD31 staining and measurement of junctions and total tubule length, HCM showed an angiogenic potential comparable to the positive control (medium with 20 ng/mL VEGF). With regard to ATE, a significantly higher number of junctions and longer tubules could be measured (vs. positive control: junctions *p* < 0.0001, total tubule length *p* < 0.01; Figure 4B,C). Counting quantitative analysis of tubular structures after addition of PL was not possible due to fibrin deposition caused by calcium in the cell culture medium (Figure 4A). As an indication of the angiogenic potential of PL, a significantly higher nitrate concentration was measured compared to the positive and negative control (Figure 4D).

## 3. Discussion

We aimed to produce and compare cell-free, protein-rich extracts from different human sources that are easily extractable and attract growth factors and cytokines into bone defects or ischemic areas for fracture healing. Recombinant growth factors are currently used in selected clinical applications [52,53]. However, they can lead to considerable side effects (e.g., inflammation, ectopic bone formation, bone resorption). They are not effective in all patients and are inferior to the gold standard of autologous bone [16,53,54,55]. Combinations of growth factors for tissue regeneration applications are potentially more efficient than the use of single ones [12,13,14,15]. To allow for a practical approach, in the present study, natural sources of growth factors were investigated. 

Several of the proteins examined by ELISA are also detectable in the fracture hematoma and could therefore play a role in improving bone formation [56]. The protein and growth factor content of ATE depends on the incubation period [31]. With a comparable incubation time of 24 h, the protein content in our study was about four times higher than reported by Sarkanen et al. (4.05 ± 0.08 mg/mL vs. 1.12 ± 0.19 mg/mL). Lopéz et al. found an almost 10 times lower mean protein concentration of 0.49 ± 0.16 mg/mL for ATE [19]. The increased protein concentration in our study may be due to an optimized production process with continuous rotation during incubation, fractional filtration and final dialysis. With PRP, Lopéz et al. described a 19-fold higher growth factor concentration compared to ATE, which is comparable to the concentration measured in our study (23-fold). To the best of our knowledge, the total protein content of HCM has not been published yet. The different protein content of the growth factor mixtures could be caused by the different production methods. During the production of PL, including repeated freeze-thaw cycles, the platelets, which contained large amounts of growth factors and cytokines in their intracellular granules, were destroyed. On the other hand, during the preparation of HCM and ATE, only a secretion of specific factors occurred with an intact cell membrane. In the literature, different concentrations of selected proteins of the growth factor mixtures are described. The most effective growth factor mixture in terms of migration and osteo- and angiogenesis remains to be determined.

As described by Lopéz et al., bFGF, VEGF, epidermal growth factor (EGF), IGF-1, IL-6, PDGF-BB, TGF-ß and TNFα are present at significantly lower concentrations in ATE compared to PRP [19]. Our study confirmed this for bFGF, VEGF, IGFBP-1 and PDGF (Table 2). EGF, tumor necrosis factor α (TNFα) and transforming growth factor β (TGF-β) were not examined by us. As determined by cytokine array and ELISA, IL-6 was detectable in HCM and ATE, but not in PL. Therefore, HCM showed the highest IL-6 concentration (804 pg/mL). This difference might be due to a high variance of the IL-6 concentration in the individual PRPs. According to Lopéz et al., 4/9 samples had very low values, whereas 5/9 had values around 350 pg/mL [19]. Gabrielyan et al. reported a VEGF concentration of 1990 pg/mL in human HCM [24]. HCM in our study had a 64–fold higher VEGF concentration (127,378 pg/mL) compared to that reported by Gabrielyan et al., leading to an improved BM-MSC migration (5% Gabrielyan et al. vs. 25% in our study) [24]. Sarkanen et al. reported an incubation-time-dependent VEGF concentration in ATE of 72.4 pg/mg protein after 24 h [31]. We observed 398 pg/mL VEGF, which corresponds to a concentration of 98.3 pg/mg protein and thus exceeds the values measured by Sarkanen et al. but is the lowest concentration in our study compared to HCM and PL. The bFGF concentrations of 264.4 pg/mg protein (equivalent to 1071 pg/mL ATE) were lower in our study. Schive et al. reported elevated levels of VEGF-A and FGF-2 in conditioned media of human adipose-derived mesenchymal stem cells after short term hypoxia and reduced levels of IL-8 and CXCL-1 [57]. Compared to PL and ATE, HCM also showed the lowest CXCL-1 level in our study. In contrast, IL-8 was lowest in PL, potentially indicating the regulation of angiogenesis in HCM from hTERT-MSCs. Antebi et al. pointed out that hypoxic culture conditions of human bone-marrow-derived MSCs suppress the IL-8 level [58]. As reported by Lozito et al., MSCs remained matrix-protective when exposed to pro-inflammatory cytokines and hypoxia, countering these stresses with increased TIMP-1 [59]. In our results, this is reflected in the highest TIMP-1 concentration of HCM.

In summary, the optimized production of the various growth factor mixtures enabled us to achieve higher protein concentrations compared with recently published data. This could be advantageous for a later clinical application. Nevertheless, even with lower total protein quantities, ATE was able to achieve inductive effects in wound healing by stimulating migration and proliferation [19]. Differences in concentration may be explained by the method of production, interindividual differences between donors, or in the use of different ELISA kits. 

In the next step, we investigated in detail the effects of these growth factor mixtures on BM-MSC migration, proliferation, osteogenic differentiation and angiogenesis in vitro. Compared to the negative control (0% FCS) PL significantly stimulated the migration of BM-MSC (*p* < 0.001), whereas HCM and ATE were less chemoattractive in our study. Gabrielyan et al. described a higher chemoattractive potential of HCM compared to pure VEGF (200 ng/mL), indicating the synergistic effects of the proteins in HCM [24]. Our results showed that proteins and cytokines are present in PL but not in HCM and ATE and stimulate cell proliferation, which is in line with the results from Herasant et al., who described an improved wound healing potential of MSCs in combination with platelet-rich plasma in mice [60]. 

PL is known to induce new bone formation [20,21]. Osugi et al. reported that serum-free conditioned media from human BM-MSC cultures enhanced the migration, proliferation and expression of osteogenic marker genes as well as new bone formation in a rat calvarial bone defect [27]. Little is known about the stimulatory potential of ATE and HCM on osteogenic differentiation. Although not proliferating when treated with HCM and ATE, we found a significantly enhanced specific ALP activity after 14 days when cultured with HCM and ATE (*p* < 0.001), which is associated with a resting period of the BM-MSC due to the lack of additional FCS. This can be an indication that HCM and ATE also have a positive influence on the formation of new bone.

With regard to the angiogenic potential, Sarkanen et al. report that a concentration of 450 µg/mL ATE is necessary to induce angiogenesis in a co-culture of HUVEC and fibroblasts [31]. Similarly, we found a significantly greater angiogenic differentiation potential compared to our positive control (20 ng/mL VEGF) in terms of junctions and total tubule length with ATE addition (4050 µg/mL). The synergistic effect of the different growth factors in ATE was predominant over the recombinant growth factor VEGF. We were able to show a comparable angiogenic potential of HCM to the positive control. This is in line with the results of Chen et al., who reported a significantly higher number of tubes, branches and a larger tubule length when seeding HUVEC onto a Matrigel matrix functionalized with conditioned medium derived from hypoxia-cultured BM-MSC compared to normoxia-cultured cells [25]. As a further indication of the angiogenic potential of HCM, Rehman et al. described a significantly increased endothelial cell growth and reduced cell apoptosis when cells were cultured with HCM from adipose-tissue-derived stromal cells (AT-MSC) [61]. This could be relevant for a clinical application of growth factors where adipose tissue of the patient could be used for ATE production, following isolation of AT-MSC and subsequent hypoxic cultivation. Even though the tubular structures could not be quantified in our study after PL addition, the angiogenic potential and the ability for endothelial cell tube formation is known from the literature and as indicated by NO measurement [62,63]. 

In summary, PL showed the highest potential concerning BM-MSC migration and proliferation, whereas HCM and ATE enhanced osteogenic and angiogenic differentiation. Thus, a combination of growth factor mixtures from different sources like platelets, adipose tissue and cell culture supernatants seems to be promising for further applications and will be investigated in detail in future studies.

A strength of this study is the direct comparison of three different naturally occurring growth factor mixtures, which were analyzed in large batches to account for biological variances and to homogenize the protein content. A further strength consists in the clinical relevance of growth factor mixtures which can be produced autologously and patient-specific and were characterized by different methods (e.g., protein analysis, cell behavior). The following limitations existed in the study design: There was no detailed investigation of mechanisms of osteogenic and angiogenic action, and the protein analysis was conducted with only of one batch per mixture. HCM was not produced from primary MSCs, and there is no verification of the findings in an animal model so far. 

## 4. Materials and Methods 

### 4.1. Generation of Growth Factor Mixtures 

#### 4.1.1. Preparation of Platelet lysates (PL)

Platelet lysates were produced from 10 expired platelet concentrates (= 25 individual donors in total) provided by the German Red Cross (BSD Ost, Dresden, Germany) as described previously [64]. Briefly, following centrifugation of the platelet concentrates (4000 × g, 10 min, 21 °C) plasma supernatant was discarded, the remaining platelet pellet was resuspended in 1/10 of the initial volume in phosphate-buffered saline (PBS) and subjected to 4 repeated freeze-thaw cycles at −80 °C/37 °C. After the 4^th^ thawing, the PL were pooled and dialyzed against ddH_2_O until electrical conductivity σ was below 10 µS/cm (dialysis tubes with cut off: 3 kDa, Scienova, Jena, Germany). Cell debris was removed by centrifugation (12,000 × g, 20 min, 21 °C) followed by sequential sterile filtration of the supernatant through 0.45 μm and 0.22 μm polyvinylidene fluoride filters (Sarstedt AG & Co. KG, Nümbrecht, Germany). Until usage, PL was stored at −80 °C. 

#### 4.1.2. Generation of Hypoxia Conditioned Medium (HCM)

Human telomerase immortalized bone-marrow-derived mesenchymal stromal cells (hTERT-MSC), kindly provided by Matthias Schieker (Laboratory of Experimental Surgery and Regenerative Medicine, University Hospital Munich (LMU), Germany) [65], were cultured in T-175 flasks in Dulbecco’s modified Eagle’s Medium (DMEM, Thermo Fisher Scientific, Waltham, MA, USA) containing 10% fetal calf serum (FCS; Sigma Aldrich, St. Louis, MO, USA) and 100 U/mL penicillin and 100 μg/mL streptomycin (Pen/Strep; both from Biochrom, Berlin, Germany) under normoxic conditions (21% O_2_, 5% CO_2_, 37 °C) until the cultures reached 95% confluence. Subsequently, the cells were rinsed once with PBS before medium was replaced by 10 mL DMEM without phenol red (Thermo Fisher Scientific, Waltham, MA, USA) containing 2% active human serum. Cultivation of the cells was done at 1% O_2_, 5% CO_2_ and 37 °C and gentle shaking for additional 5 days. Thereafter, HCM was collected, dialyzed against ddH_2_O until σ < 10 µS/cm (dialysis tubes with cut off: 3 kDa, Scienova, Jena, Germany), pooled from 2 batches and stored at −80 °C until further use.

#### 4.1.3. Preparation of Adipose Tissue Extracts (ATE)

After approval of the Institutional Review Board of Technische Universität Dresden (project identification code: EK 360092014; 2014/10/09) adipose tissue was harvested after informed consent from 5 healthy human donors undergoing reconstructive surgery, and ATE was prepared according to a modified protocol by Sarkanen et al. [31]. Briefly, adipose tissue specimens were cut into large pieces and transferred into 50 mL tubes. Equal volumes of DMEM without phenol red, serum and antibiotics were added into the tubes followed by incubation at 37 °C in a rotating device for 24 h for growth factor extraction. After incubation adipose tissue was removed, the remaining medium (= ATE) was collected, filtered through a 40 μm filter and centrifuged (2000 rpm, 10 min). Supernatants were dialyzed against ddH_2_O until σ < 10 µS/cm (dialysis tubes with cut off: 3 kDa, Scienova, Jena, Germany), pooled, filtered through a 0.22 μm filter, aliquoted and stored at −80 °C until use.

### 4.2. Biochemical Characterization of Growth Factor Mixtures

#### 4.2.1. Protein Quantification Assay

PL, HCM and ATE samples were analyzed for the total protein content according to the manufacturer’s instructions for Bradford assay (Roti^®^-Quant, Carl Roth GmbH & Co. KG, Karlsruhe, Germany). Absorbance measurements were performed at 595 nm (Infinite^®^ M200 Pro, Tecan, Männedorf, Switzerland). 

#### 4.2.2. Angiogenesis and Cytokine Array

Proteome profiler arrays for 36 human cyto- and chemokines and 55 human angiogenesis-related proteins were performed according to the manufacturer’s instructions (Proteome Profiler Human Angiogenesis Array Kit and Proteome Profiler Human Cytokine Array Kit, R&D, Minneapolis, MN, USA). Chemiluminescence signals were detected with a gel documentation system (G:BOX, Syngene, Cambrige, UK) and analyzed qualitatively and quantitatively using ImageJ software (National Institutes of Health, Bethesda, MD, USA). 

#### 4.2.3. Enzyme-Linked Immunosorbent Assay

In order to quantify the content of selected proteins, enzyme-linked immunosorbent assays (ELISA) were performed according to the manufacturer’s instructions. Tissue inhibitor of metalloproteinases 1 (TIMP-1), VEGF, platelet-derived growth factor-BB (PDGF-BB), basic fibroblast growth factor (bFGF), interleukin 6 (IL-6), interleukin 8 (IL-8), interleukin 1ß (IL-1ß) and interleukin 10 (IL-10) ELISA kits were from PeproTech GmbH (Hamburg, Germany), angiogenin and insulin-like growth factor-binding protein 1 (IGFBP-1) kits were from Sigma-Aldrich (St. Louis, MO, USA) and the chemokine (C-X-C motif) ligand 1 (CXCL-1/GROα) kit was from R&D (Minneapolis, MN, USA). Absorbance measurements were performed according to the manufacturer’s specifications (Infinite^®^ M200 Pro, Tecan, Männedorf, Switzerland).

### 4.3. Cell Culture Experiments to Investigate the Effects of Growth Factor Mixtures

#### 4.3.1. Cells

Primary human BM-MSC were isolated from bone marrow aspirates of 4 donors (3 male, age 24–33; 1 female, age 60; all Caucasian), expanded in α-MEM/GlutaMAX containing 15% FCS and Pen/Strep and used in passage 3 and 4 for the experiments. The use of BM-MSC was approved by the ethics commission of the Technische Universität Dresden.

Human umbilical vein endothelial cells (HUVEC) were purchased from Promocell (Germany), cultivated in Endothelial Cell Growth Medium (Promocell, Heidelberg, Germany) and used in passage 4 for the experiments.

#### 4.3.2. Chemotaxis Assay

The chemoattractive potential of the growth factor mixtures was tested using a transwell migration assay (Corning^®^ HTS Transwell^®^-96 permeable supports, Sigma Aldrich, St. Louis, MO, USA) with a pore size of 8.0 μm. 2.5 × 10^4^ primary human BM-MSC, that had been starved in serum-free medium for 24 h and seeded in 75 µL DMEM without phenol red or other supplements into the upper chamber, whereas 150 µL of the different growth factor mixtures (PL, HCM, ATE) or DMEM with 0% FCS (negative control) and 30% FCS (positive control), respectively, were added into the lower chamber as chemoattractant. After 24 h, medium was removed, cells were washed with PBS and non-migrated cells on top of the membrane were removed with a cotton swab. The number of migrated cells was determined by measurement of lactate dehydrogenase (LDH) activity after cell lysis (see Section 4.3.4). Experiments were performed in triplicates with 4 different BM-MSC donors.

#### 4.3.3. Osteogenic Differentiation of BM-MSC

Primary human BM-MSC were seeded at 1 × 10^4^ cells/cm^2^ in cell culture plates and cultivated for up to 14 days with DMEM containing Pen/Strep, 10 IU/mL heparin (Rotexmedica GmbH, Trittau, Germany), 100 nM dexamethasone (Dex), 50 μM L-ascorbic acid 2-phosphate (AAP), 10 mM ß-glycerol phosphate (ß-GP; all from Sigma Aldrich, St. Louis, MO, USA) and either 10% FCS (positive control) or 10% PL, HCM or ATE. Medium with 10% FCS but without Dex, AAP or ß-GP served as negative control. After 3 and 14 days, cells were washed twice with PBS and frozen at −80 °C until LDH and alkaline phosphatase (ALP) activity measurement. Experiments were performed in triplicates with 2 different BM-MSC donors.

#### 4.3.4. Analysis of LDH and ALP Activity

Frozen samples were thawed on ice for 20 min following incubation with 1% Triton-X-100/PBS (Sigma Aldrich, St. Louis, MO, USA) on ice for 50 min. LDH activity was determined using the CytoTox 96^®^ Non-Radioactive Cytotoxicity Assay (Promega, Madison, WI, USA) according to manufacturer’s instructions by measuring the absorbance at 490 nm (Infinite^®^ M200 Pro). Cell number was calculated from LDH activity of defined cell numbers.

ALP activity was determined by incubating an aliquot of the same cell lysate used for LDH quantification with 1 mg/mL p-nitrophenyl phosphate (Sigma Aldrich) in 0.1 M diethanolamine, 1% Triton X-100, 1 mM MgCl_2_ (pH 9.8). After incubation for 30 min at 37 °C, the enzymatic reaction was stopped by adding 1 M NaOH, and absorbance was measured at 405 nm (Infinite^®^ M200 Pro, Tecan, Männedorf, Switzerland). Specific ALP activity was calculated by correlating the absorbance to a p-nitrophenol calibration line and the respective cell number.

#### 4.3.5. In vitro Angiogenesis Assay

6.8 × 10^3^ BM-MSC were seeded into 96-well plates and cultured for 2 days in α-MEM containing 15% FCS and Pen/Strep before seeding of 1.7 × 10^3^ HUVEC on top of the BM-MSC monolayer (ratio BM-MSC:HUVEC = 4:1) [66]. A 1:1 mixture of α-MEM and Endothelial Cell Basal Medium (Promocell, Heidelberg, Germany) containing 10% heat-inactivated FCS, Pen/Strep and osteogenic supplements (100 nM Dex, 50 µM AAP, 5 mM β-GP) was used as co-culture medium (= negative control). For investigating the effect of the different growth factor mixtures on angiogenesis PL, HCM and ATE were lyophilized, reconstituted in ddH_2_O to 1/10 of the initial volume and added to the co-culture medium at 10%. Co-culture medium with 20 ng/µL VEGF_165_ (PeproTech GmbH, Hamburg, Germany) served as positive control. After 10 days of co-culture, medium supernatants were collected and frozen at −80 °C until nitrate measurement (see Section 4.3.7). The cells were washed with PBS and fixed in 4% neutral buffered formaldehyde (SAV LP GmbH, Flintsbach am Inn, Germany).

#### 4.3.6. CD31-Staining and Analysis of Angiogenic Structures

To visualize the formed tubular structures, fixed samples were washed 3x with PBS, incubated for 30 min with 5% goat serum to block unspecific binding sites, washed 3x with PBS again and incubated for 1 h with a monoclonal antibody against CD31 (mouse anti-human, 1:200 in PBS; DAKO, Santa Clara, CA, USA). After 3x washing with PBS, ZytoChem Plus HRP One-Step Polymer anti-Mouse/Rabbit/Rat (ready-to-use; Zytomed Systems GmbH, Bargteheide, Germany) was added for 30 min. Then, samples were washed 3x with PBS and incubated with DAB (3,3’-diaminobenzidine; Zytomed Systems GmbH, Bargteheide, Germany) for approximately 5–10 min. Widefield microscopy was performed using a BZ-9000 microscope (Keyence, Osaka, Japan), and images were analyzed using ImageJ 1.48t (Wayne Rasband, Kensington, MD, USA) and Angiosys 1.0 (TCS Cellworks, Buckingham, UK).

#### 4.3.7. Nitrate Measurement

In order to quantify the nitric oxide (NO) production, which plays an important role e.g., in vascular regulation, nitrate as the oxidation product was measured in medium supernatants collected at the end of the angiogenesis co-culture experiment using the Nitric Oxide Colorimetric Assay (BioVision, Milpitas, CA, USA) according to manufacturer’s instructions. Absorbance measurements were performed according to the manufacturer’s specifications (Infinite^®^ M200 Pro, Tecan, Männedorf, Switzerland).

### 4.4. Statistical Analysis

For multiple comparisons, statistical analysis was performed with one-way ANOVA following Bonferroni post-hoc testing by using GraphPad PRISM 7 (Graphpad Software, San Diego, CA, USA). Significance levels were set as * *p* < 0.05, ** *p* < 0.01 and *** *p* < 0.001. Data are presented as mean ± standard deviation (SD).

## 5. Conclusions

Multiple angiogenic proteins and cytokines were detected in all three tested naturally occurring growth factor mixtures independently of their different total protein content. To combine the chemoattractive potential with osteogenic and angiogenic differentiation capacity, a combination of different growth factors might be beneficial. Scaffold functionalization with concentrated growth factor mixtures appears attractive for a later clinical application, and release experiments are needed.

## Figures and Tables

**Figure 1 ijms-21-01412-f001:**
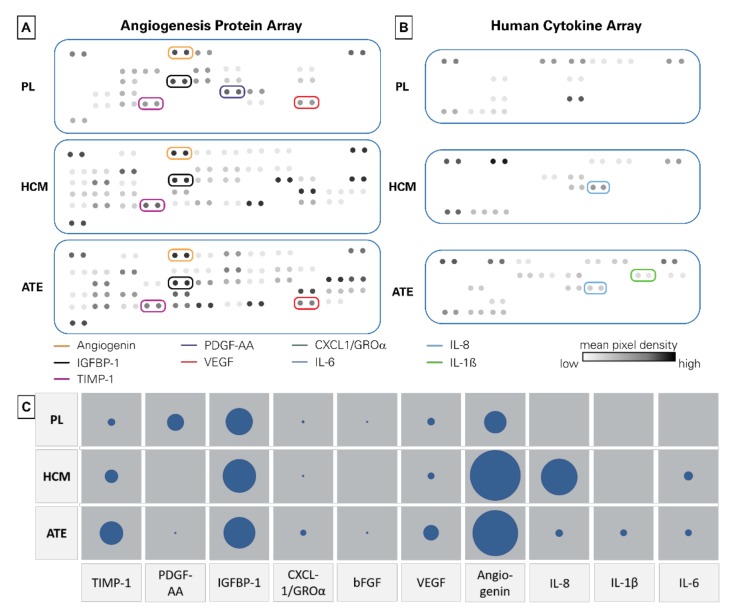
Protein and cytokine array data. (**A**) Angiogenesis protein array profile, (**B**) cytokine array profile and (**C**) mean pixel density of selected proteins plotted as circles (the larger the circle, the higher the mean pixel density).

**Figure 2 ijms-21-01412-f002:**
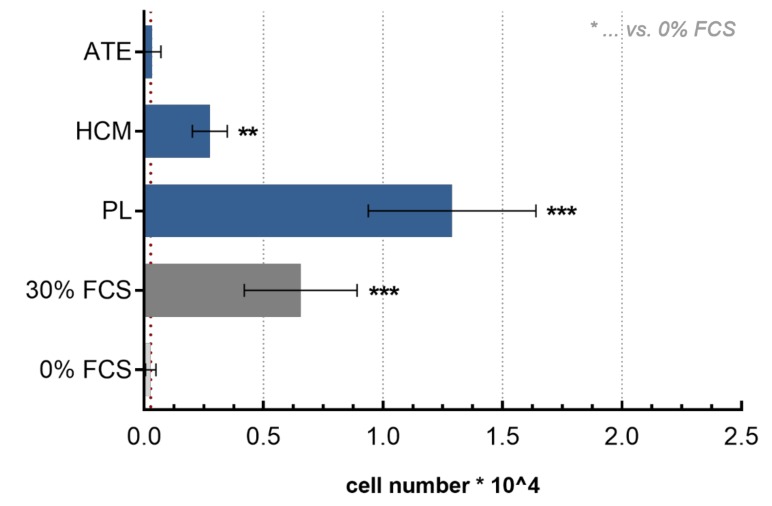
Transwell migration assay. Number of migrated bone-marrow-derived mesenchymal stem cells (BM-MSC) after 24 h incubation as determined by lactate dehydrogenase (LDH) activity measurement (mean ± SD, *n* = 12; ** *p* < 0.01, *** *p* < 0.001).

**Figure 3 ijms-21-01412-f003:**
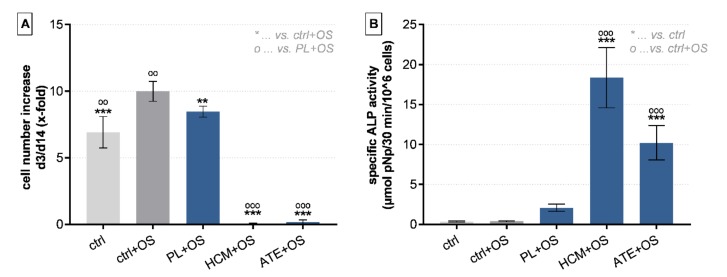
Effect of growth factor mixtures on proliferation and osteogenic differentiation of BM-MSCs. (**A**) Cell number increase from day 3 to day 14 and (**B**) specific ALP activity after 14 days of cultivation (mean ± SD, *n* = 6; **/°° *p* < 0.01, ***/°°° *p* < 0.001). OS: osteogenic supplements.

**Figure 4 ijms-21-01412-f004:**
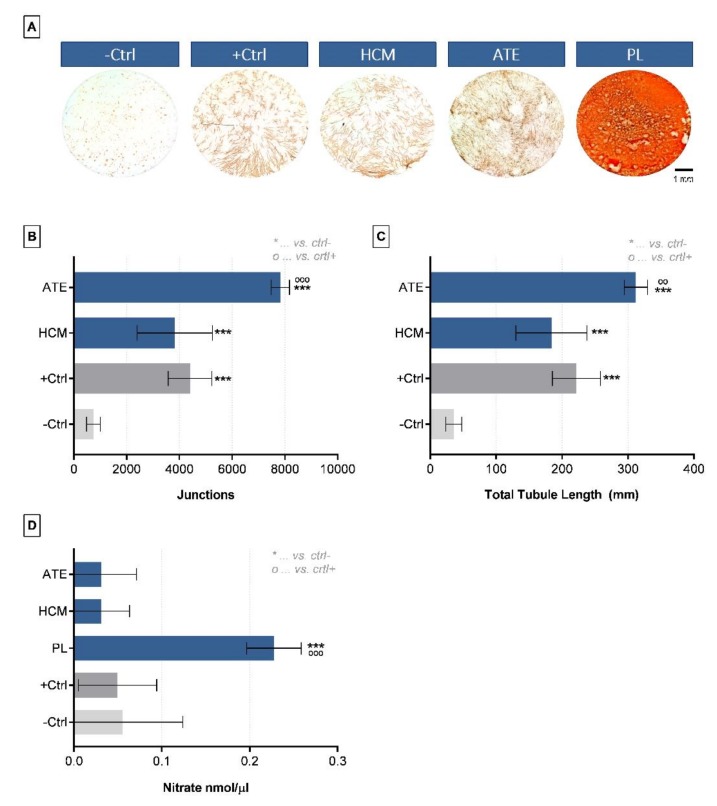
Angiogenic potential of growth factor mixtures as analyzed by a co-culture of osteogenically induced BM-MSCs and HUVECs. (**A**) Light microscopic images of tubular structures (visualized by CD31 immunostaining) after 10 days of cultivation in co-culture medium with different growth factor mixtures, (**B**) number of junctions, (**C**) total tubule length and (**D**) nitrate concentration (mean ± SD, *n* = 3; °° *p* < 0.01, ***/°°° *p* < 0.001). -Ctrl: only co-culture medium, +Ctrl: co-culture medium + 20 ng/mL VEGF.

**Table 1 ijms-21-01412-t001:** Total protein content of the different growth factor mixtures (*n* = 3).

	Mean ± SD
**PL**	11.60 ± 0.15 mg/mL
**HCM**	0.015 ± 0.006 mg/mL
**ATE**	4.05 ± 0.08 mg/mL

**Table 2 ijms-21-01412-t002:** ELISA results of selected proteins of bioactive factors derived from adipose tissue (ATE), platelet lysate (PL) and conditioned medium from hypoxia-treated immortalized bone-marrow-derived mesenchymal stromal cells (HCM).

	PL (pg/mL)	HCM (pg/mL)	ATE (pg/mL)	Function (Selected)
TIMP-1	498	1992	924	angiogenesis suppression [33,34]
PDGF	8233	0	60	blood vessel formation, proliferation [35]
IGFBP-1	2905	3026	1417	cell migration and metabolism [36]
CXCL1 (GROα)	1970	1469	2721	activates neutrophils and basophils; increases cell migration [37,38]
bFGF	1721	708	1071	cell migration, activated during wound healing [39]
VEGF	605		398	stimulates the formation of blood vessels [40,41,42]
Angiogenin	159	1447	890	cell migration, proliferation and formation of tubular structures [43,44]
IL-8	22	1105	2112	innate immune response, regulated angiogenesis [45,46]
IL-1β	30	3	124	inflammatory response, cellular activities–proliferation and differentiation [47,48]
IL-10	135	15	201	immunoregulation and inflammation [49]
IL-6	0	804	784	pro- and anti-inflammatory properties [50,51]

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
