# Peer review of "Characterization of Naturally Occurring Bioactive Factor Mixtures for Bone Regeneration"

_ijms, 2020, doi:10.3390/ijms21041412_

Round 1

Reviewer 1 Report

Summary: The authors of the manuscript reported on the bone regeneration and angiogenic potential of naturally-derived  bioactive factors from adipose tissue extract (ATE), platelet lysate from PRP (PL) and conditioned medium from hypoxic hTERT immortalized human bone marrow-derived mesenchymal stem cells (hTERT-MSC). To this purpose, they obtained these biological samples and used them to treat primary human BM-MSC in monoculture or co-culture with HUVEC. There have been investigated, comparatively, the protein concentration and the expression pattern of the bioactive molecules contained by these three biological samples and their chemotactic, mitogenic, osteogenic and adipogenic effects. The authors point out the clinical applications for the bone regeneration of the synergism between chemoattractant, osteogenic, and angiogenic capacities of naturally-derived bioactive factor mixtures.

You will find hereinafter my comments and suggestions:

1. “Abstract” covers the main elements of the manuscript and “Introduction” section provides the context for the approached topic. 

2.The methods performed are clearly described to allow the interested readers to reproduce them.

3. I have only a few concerns regarding the followings:

a) The authors stated “Angiogenin was the most abundant protein in HCM and ATE, whereas in PL amounts of IGFBP-1 were highest” (lines 82-83). Please revise as IGFBP1 amounts were highest in case of HCM although not significantly higher than PL. Please also analyse the next statements within this paragraph for accuracy.

b) The authors wrote “Our study confirmed this for bFGF, VEGF, IGFBP-1, PDGF-BB and tumor necrosis factor α (TNFα)” (line 165). Please revise since TNFα was not examined/presented in this study.

c) It would be great if the authors could better discuss on connections between the levels of the bioactive molecules detected in the three biological samples and the effects on the evinced cellular responses in terms of cell migration, proliferation, and osteogenic and adipogenic potential.

4. Minor comments:

a) Please expand completely the abbreviation hTERT-MSC.

b) Please check carefully the „References” section since the titles of some journals are not abbreviated. For instance, Refs. 21, 30, 32, 37, 39….

9….

Author Response

Comments and Suggestions for Authors

Summary: The authors of the manuscript reported on the bone regeneration and angiogenic potential of naturally-derived bioactive factors from adipose tissue extract (ATE), platelet lysate from PRP (PL) and conditioned medium from hypoxic hTERT immortalized human bone marrow-derived mesenchymal stem cells (hTERT-MSC). To this purpose, they obtained these biological samples and used them to treat primary human BM-MSC in monoculture or co-culture with HUVEC. There have been investigated, comparatively, the protein concentration and the expression pattern of the bioactive molecules contained by these three biological samples and their chemotactic, mitogenic, osteogenic and adipogenic effects. The authors point out the clinical applications for the bone regeneration of the synergism between chemoattractant, osteogenic, and angiogenic capacities of naturally-derived bioactive factor mixtures.

You will find hereinafter my comments and suggestions:

“Abstract” covers the main elements of the manuscript and “Introduction” section provides the context for the approached topic. 

2.The methods performed are clearly described to allow the interested readers to reproduce them.

I have only a few concerns regarding the followings:

a) The authors stated “Angiogenin was the most abundant protein in HCM and ATE, whereas in PL amounts of IGFBP-1 were highest” (lines 82-83). Please revise as IGFBP1 amounts were highest in case of HCM although not significantly higher than PL. Please also analyse the next statements within this paragraph for accuracy.

Answer:

In the text passage you mentioned we refer to the protein array data, not to the ELISA data. To make it clear we modified the text to: “With respect to the protein array analysis, Angiogenin was the most abundant protein in HCM and ATE, whereas in PL amounts of IGFBP-1 were highest.”

Additionally, we will change the order of the tables/figures, showing table 1 (total protein content) firstly, figure 1 (protein array data) secondly and table 3 (ELISA data) thirdly.

b) The authors wrote “Our study confirmed this for bFGF, VEGF, IGFBP-1, PDGF-AA and tumor necrosis factor α (TNFα)” (line 165). Please revise since TNFα was not examined/presented in this study.

Answer:

Thanks for this hint. TNF-alpha was removed from the statement.  

c) It would be great if the authors could better discuss on connections between the levels of the bioactive molecules detected in the three biological samples and the effects on the evinced cellular responses in terms of cell migration, proliferation, and osteogenic and adipogenic potential.

Thank you for this comment. We have added the discussion regarding the ELISA results (line 182-189).

Minor comments:

a) Please expand completely the abbreviation hTERT-MSC.

Answer:

Thank you for this suggestion, we will expand hTERT-MSC to human telomerase immortalized bone marrow-derived mesenchymal stromal cells.

b) Please check carefully the „References” section since the titles of some journals are not abbreviated. For instance, Refs. 21, 30, 32, 37, 39….

Answer:

We have checked the reference section and we abbreviated all journal titles.

Reviewer 2 Report

The authors compared the growth factors mixtures from adipose tissue extract (ATE), conditioned media from hypoxic mesenchymal stem cells (HMC) and platelet lysates (PL) in promoting osteogenesis and angiogenesis of bone marrow mesenchymal stem cells. They showed that PL was most potent in inducing these effects. Overall, the manuscript is easy to understand and well-structured. I only have a few comments. Why the ELISA results did not have SD values? Were the protein concentrations measured only once in one batch of samples? The authors need to mention the limitations of the study, including that the osteogenic and angiogenic mechanisms of each product were not tested, and the need to verify the findings in animal studies. In the last line of discussion, the authors concluded that a combination of growth factors from different sources seems to be promising (for angiogenic and osteogenic effects). By different sources, I assume from ATE. HMC and PL. However, this was not tested at all. At most, they showed that PL is more potent than ATE and HMU. The authors may need to modify the concluding statement.

Author Response

Comments and Suggestions for Authors

The authors compared the growth factors mixtures from adipose tissue extract (ATE), conditioned media from hypoxic mesenchymal stem cells (HMC) and platelet lysates (PL) in promoting osteogenesis and angiogenesis of bone marrow mesenchymal stem cells. They showed that PL was most potent in inducing these effects. Overall, the manuscript is easy to understand and well-structured.

I only have a few comments.

Why the ELISA results did not have SD values? Were the protein concentrations measured only once in one batch of samples?

Answer:

Large batches of the different growth factor mixtures were prepared as described in the materials and methods sections. For the ELISA measurements only one sample out of the batch was measured concerning protein concentration why there are no SD values.

The authors need to mention the limitations of the study, including that the osteogenic and angiogenic mechanisms of each product were not tested, and the need to verify the findings in animal studies.

Answer:

Thank you for this comment. Limitations were added in the manuscript.

Strength:

Direct comparison of 3 different naturally occurring growth factor mixtures Large batches of mixtures to account for biological variances and to homogenize protein content Growth factor mixtures clinically relevant, autologous patient-individual mixtures producible Characterization of mixtures by different methods (protein analysis, cell behavior)

Limitations:

No detailed investigation of mechanisms of osteogenic and angiogenic action Protein analysis only of 1 batch per mixture HCM not from primary MSCs So far no verification of findings in animal model

Thank you for this advice. It is right that the detailed mechanisms of action regarding osteogenic and angiogenic potential of each product were not investigated and that results have to be verified in animal studies. We will mention the strength and limitations of this study in the discussion section.

In the last line of discussion, the authors concluded that a combination of growth factors from different sources seems to be promising (for angiogenic and osteogenic effects). By different sources, I assume from ATE. HMC and PL. However, this was not tested at all. At most, they showed that PL is more potent than ATE and HMU. The authors may need to modify the concluding statement.

Answer:

Thank you for this hint. The combination of PL, HCM and ATE will be tested in following studies. We revised the last concluding statement to “Thus, a combination of growth factor mixtures from different sources like platelets, adipose tissue, and cell culture supernatants seems to be promising for further applications and will be investigated in detail in following studies.

Reviewer 3 Report

This study compared bioactive factors release and angiogenic properties among PL, HCM and ATE experimental groups in vitro. The idea is interesting and this manuscript is well prepared. However, some descriptions and viewpoints should be clarified.

Overall, how to convince or decide the amount or dose (weight or volume) from PL, HCM and ATE experimental groups is similar in cell culture experiments? This is because the amount or dose of growth factor mixtures from these three groups will influence the results. In addition, how to decide the exact amounts of growth factor mixtures in further experiments. In the section 2.1, do the measurements of Angiogenesis Protein and Cytokine Array from only one sample? It will be much better from three samples to avoid the individual differences.   In the section 2.1, why using protein assays and ELISA assays at the same time? Which assay is better to reveal the exactly angiogenic and immunologic properties of growth factor mixtures? This is because some controversies occurred between these two analyses.    In the section 2.2. and Figure 3A, no cell number increase from day 3 to day 14 is found in HCM+OS and ATE+OS experimental groups. Does this finding demonstrate cell toxicity when exposing to HCM and ATE? Or the figure cause readers misunderstanding. Thus, it is better to adjust the figure to reveal more information.      In the section 2.3. and Figure 4B/C, why the PL group did not show in the Figure 4B and 4C. Moreover, osteogenic differentiation using Alizarin Red S staining should be added in this experiment to demonstrate the osteogenic properties of these three experimental groups.     Line 166: “EGF and transforming growth factor β (TGF-β) and were not examined by us.” “and” should be deleted. Line 190, this manuscript only compared ALP activity among these three experimental groups. Please extend more experiments to demonstrate the differences of osteogenic properties among experimental groups

Author Response

Comments and Suggestions for Authors

This study compared bioactive factors release and angiogenic properties among PL, HCM and ATE experimental groups in vitro. The idea is interesting and this manuscript is well prepared. However, some descriptions and viewpoints should be clarified.

Overall, how to convince or decide the amount or dose (weight or volume) from PL, HCM and ATE experimental groups is similar in cell culture experiments? This is because the amount or dose of growth factor mixtures from these three groups will influence the results.

Answer:

You are absolutely right that some kind of normalization is needed to compare the results. To account for biological variances for all growth factor mixtures large batches were prepared. All experiments were performed with this one batch and with the same concentrations (vol.%) in the cell culture experiments.

In addition, how to decide the exact amounts of growth factor mixtures in further experiments. In the section 2.1, do the measurements of Angiogenesis Protein and Cytokine Array from only one sample? It will be much better from three samples to avoid the individual differences.  

Answer:

Thanks for this comment. As mentioned above, to account for biological variances PL and ATE were prepared from different donors and pooled at the end to create a mostly homogeneous batch (PL: 25 individual donors, ATE: 5 individual donors). For production of HCM a human telomerase immortalized bone marrow-derived mesenchymal stromal cell line was used whose secretome shows nearly no variances as in contrast to primary cells. Due to these large homogeneous batches only one sample out of every batch was used for the protein array and ELISA measurements. In addition, all experiments were performed with this batch. For any further experiments using a new batch of growth factor mixtures or even mixtures prepared from individual donors, exact amounts of growth factors have to be determined again to compare the results.

In the section 2.1, why using protein assays and ELISA assays at the same time? Which assay is better to reveal the exactly angiogenic and immunologic properties of growth factor mixtures? This is because some controversies occurred between these two analyses.   

Answer:

Thank you for this question. When producing the growth factor mixtures, they are like a big black box. To have an idea which proteins and cytokines are present in the mixtures, as a first step we performed the arrays. But they give only a semi-quantitative result (present/not present, density of pixels). Based on the array results we have chosen some proteins and cytokines for quantitative analysis by ELISA. In our opinion, these protein/cytokine arrays are a nice tool to get an impression of what is inside such unknown mixtures. But they cannot replace exact measurements as they are necessary to determine the biologically relevant/effective dose for cell culture and especially in vivo experiments.

In the section 2.2. and Figure 3A, no cell number increase from day 3 to day 14 is found in HCM+OS and ATE+OS experimental groups. Does this finding demonstrate cell toxicity when exposing to HCM and ATE? Or the figure cause readers misunderstanding. Thus, it is better to adjust the figure to reveal more information.     

Answer:

Thanks for this valuable comment. For this experiment cells were cultured in DMEM with 1 % antibiotics and osteogenic supplements. Depending on the experimental group either 10 % FCS (positive control) or 10 % PL, HCM or ATE were additionally added.

For sufficient proliferation MSCs will need a cocktail of proteins as present for example in FCS. Our results showed that proteins and cytokines present in PL, but not that present in HCM and ATE stimulate cell proliferation, not meaning that HCM and ATE are toxic. For better understanding it would have been better to include also a negative control (DMEM with 1 % antibiotics ± osteogenic supplements but w/o 10 % FCS) to show growth arrest when FCS is missing.

On the other hand, it is well known that proliferation and differentiation are controversy processes that could be shown in our study: although not proliferating when treated with HCM and ATE, cells differentiated into the osteogenic lineage as verified by a high cell-specific ALP activity.

We will address this issue in the discussion section to avoid any misunderstandings.

In the section 2.3. and Figure 4B/C, why the PL group did not show in the Figure 4B and 4C.

Answer:

When performing the angiogenesis assay the fibrinogen present in the PL reacted with the calcium present in the cell culture medium leading to fibrin depositions. After staining a reddish fibrin film laying on the cells was visible (see Figure 4A) not allowing counting of tubular structures. Therefore, no data for quantitative analysis are available for PL.

As an indication of the angiogenic potential of PL, nitrate concentration was measured and compared to HCM, ATE as well as to the positive and negative control (Figure 4D).

Moreover, osteogenic differentiation using Alizarin Red S staining should be added in this experiment to demonstrate the osteogenic properties of these three experimental groups.   

Answer:

Thanks for this comment. We determined the specific ALP activity as an early marker for osteogenic differentiation. You are right that to gain deeper insights into the osteogenic differentiation process mineralization should be verified by e. g. Alizarin Red S staining. We will keep this in mind and implement this assay in our future investigations.

Line 166: “EGF and transforming growth factor β (TGF-β) and were not examined by us.” “and” should be deleted.

Answer:

Thanks a lot for this hint - “and” will be deleted in the respective line.

Line 190, this manuscript only compared ALP activity among these three experimental groups. Please extend more experiments to demonstrate the differences of osteogenic properties among experimental groups.

Answer:

Thanks for this comment. At this time our study focused on the production and characterization of PL, HCM and ATE. The 3 groups were compared by performing only selected experiments to gain a first impression of the behavior of these mixtures. We will include more experiments demonstrating differences of osteogenic properties in our future experiments.

Round 2

Reviewer 1 Report

The revised manuscript deserves to be published in Int J Mol Sci journal.

Reviewer 3 Report

Accept